# A Longitudinal Investigation on the Effects of Sodium and Potassium Intake on the Development of Hypertension and Abdominal Obesity from Childhood to Young Adulthood amongst Ellisras Rural Population, South Africa

**DOI:** 10.3390/children10081330

**Published:** 2023-08-01

**Authors:** Thato Tshepo Raphadu, Peter Modupi Mphekgwana, Moloko Matshipi, Kotsedi Daniel Monyeki

**Affiliations:** 1Department of Physiology and Environmental Health, University of Limpopo, Sovenga 0727, South Africa; thatoraphadu@gmail.com (T.T.R.); moloko.matshipi@ul.ac.za (M.M.); 2Department of Research Administration and Development, University of Limpopo, Sovenga 0727, South Africa; peter.mphekgwana@ul.ac.za

**Keywords:** hypertension, abdominal obesity, longitudinal study

## Abstract

Background: Hypertension (HT) and obesity have both been on the rise in children. Each is associated with an increase in cardiovascular disease risk, and both track into adulthood. Objectives: This study aimed to identify the association of sodium intake (Na), potassium (K) intake, and sodium-to-potassium (Na/K) ratio with the development of HT and abdominal obesity amongst the Ellisras rural population over time. Methods: In this longitudinal study, data on dietary intake of Na and K were collected using a 24 h recall questionnaire from a total of 325 participants tracked from 1999 (5–12 years), 2001 (7–14 years), and 2015 (18–30 years). The averages of Na and K intake were analysed using local food tables and the South African Food Composition Database System (SAFOODS). In addition, blood pressure (BP) and anthropometric measurements (waist circumference (WC) and height) were also examined. Parametric (independent *t*-test) and Chi-square/Fishers exact tests were conducted to determine the difference between the years for numerical data and categorical variables. A generalised estimating equation (GEE) was used to assess the association of Na intake, K intake and their ratio on BP, WC, and waist-to-height ratio (WHtR). Results: Our results indicate a significant positive association between K intake and WHtR, and even though the model was adjusted for age and sex, there was still an association with WHtR. The Na/K ratio was associated with both BP measurements and abdominal obesity, respectively. Furthermore, Na/K was shown to be associated with an increased risk of developing HT and abdominal obesity. Conclusions: In our study, we observed that an increase in the Na/K ratio is a predictor of HT and abdominal obesity over time compared to Na and K alone. However, more studies are required to further prove this.

## 1. Introduction

Hypertension and obesity have both been on the rise in children. Both track into adulthood and are linked to an increased risk of cardiovascular disease, which increases the prevalence of heart disease and its associated morbidity and mortality [1]. South Africa is an upper-middle-income country in which hypertension is a highly prevalent condition. In South Africa, hypertension was accountable for almost 47,000 fatalities in 2000. Since then, the prevalence has increased from 25% to more than 40% of the population [2]. According to a 2019 government study, around 41% of women and 11% of males aged 15 and above were obese [3]. Dietary intake has been identified as one of the most important risk factors for non-communicable diseases. High sodium and low potassium intakes are associated with the early development of chronic diseases (e.g., hypertension and obesity) [4].

Potassium is an essential mineral for human health [5,6]. It is essential for the normal functioning of cells and organs [7,8] due to its involvement in nerve transmission, muscle contractions, blood pressure regulation, and skeleton integrity [5,9,10,11]. Fruits, legumes, starchy roots and tubers, whole grains, and vegetables are the primary dietary sources of potassium [6,12,13]. The regulation of urinary potassium excretion is primarily responsible for the homeostatic control of serum potassium concentration. Reduced dietary intake of potassium [6], increased consumption of sodium [14,15] due to food processing, and high salt intake all have an effect on potassium utilization [16]. Dietary potassium intake is reduced due to potassium loss during food processing [17].

Evidence reveals a link between sodium consumption and blood pressure (BP) levels [18]. Excess sodium consumption has been associated with the development of hypertension and its cardiovascular consequences [19]. Reduced sodium consumption, on the other hand, not only lowers blood pressure and hypertension incidence but also lowers cardiovascular morbidity and death [20]. Observational studies, on the other hand, show that dietary potassium intake is inversely linked with blood pressure in certain studies but not in others [21]. It is known that sodium and potassium can affect blood pressure through several mechanisms. According to evidence, these micronutrients interact in a way that significantly contributes to the development of primary hypertension [22,23]. Modern Western diets—which are particularly heavy in sodium and low in potassium—have a biological interaction with the kidneys that causes the body to have excessive sodium levels and insufficient potassium concentrations. Vascular smooth muscle cell concentration is the first biological effect of these alterations, which are then followed by an increase in peripheral vascular resistance, increased pressure, and hypertension [22,23].

Obesity has lately attracted attention as another potential health-related result despite the distinct detrimental effects of increased sodium intake on hypertension [24]. Adult obesity and sodium intake are significantly correlated, according to several studies [25]. Furthermore, research from nationally representative populations in Australia, South Korea, and the UK revealed a substantial link between sodium and obesity in both adults and children [26]. The role of soft drink intake was investigated, despite the fact that the mechanisms underlying the link between sodium and obesity have not yet been thoroughly established [27]. According to this hypothesis, higher energy intake from sugar-sweetened beverages used to quench the thirst brought on by high sodium intake would contribute to the development of obesity [28]. In other words, salt intake is thought to be indirectly related to soft drink consumption, which in turn is related to obesity due to its impact on fluid intake [29]. Although there is sufficient evidence regarding Na and K, there are very few longitudinal studies in Africa studying the effect of sodium and potassium intake and their ratio on hypertension and abdominal obesity from childhood into young adulthood. Most of the studies were conducted in Europe, Asia, and North America. A study conducted in Africa by Saeid et al. 2018 only assessed the proportion frequency of Na and K [30] and did not track the changes in Na, K and NA/K ratio over time. The study conducted in Ellisras rural children by van Den Ende et al. (2014) and Mashiane et al. (2018) focused their study on the association between Body Mass Index (BMI) and dietary intake in children and young adults, respectively [31,32]. Studies regarding the association between micronutrients such as Na, K, and hypertension and abdominal obesity have received little attention, and the relationship between dietary intake and hypertension and abdominal obesity is inconsistent. This discrepancy highlights the need for such a study, especially in rural African populations, and the need to provide data from an African perspective and not only from Western and Eastern countries. Therefore, we designed this study to investigate the effect of sodium and potassium intake and their ratio on hypertension and abdominal obesity from childhood into young adulthood in Ellisras.

## 2. Materials and Methods

### 2.1. Sampling Procedure

This study is a component of the Ellisras Longitudinal Study (ELS), the regional specifics of which have been covered elsewhere [33]. At first, the ELS used a cluster sampling strategy. Basically, 22 schools were randomly selected from 68 schools in the Ellisras rural region to participate in the study (10 preschool and 12 primary schools) [33]. Aged 3 to 10 years, baseline data were gathered in 1996. Subsequent examinations of dietary consumption, blood pressure, and anthropometric measures were performed in 1999, 2001, 2015, and 2018. Longitudinal data from 1999, 2001, and 2015 were analysed for this study. The ultimate number of participants was 325, of whom 164 women and 161 men had complete measurements (Figure 1).

### 2.2. Blood Pressure and Anthropometric Measurements

As described by Whelton et al. (2018) [34], blood pressure was monitored using Omron M2 blood pressure equipment. The National Heart, Lung, and Blood Institute (NHLBI) (2005) established cut-off points for high blood pressure or hypertension for age, sex, and height [35]. Boys’ and girls’ blood pressure was determined individually using the Merck manual medical calculator [36]. In contrast to adults, this method of computation calculates blood pressure for children and adolescents using percentiles. For adolescents over the age of eighteen and young adults, the cut-off points recommendations outlined by Whelton et al. (2018) were applied [34]. Weight, height, and waist circumference were measured anthropometrically in accordance with the International Society for the Advancement of Kinanthropometry’s recommended practices [37]. The WC was divided by height in centimetres to obtain the waist-to-height ratio. Waist circumference (≥90th percentile for children and adolescents, ≥90 cm for males and ≥80 cm for females) and waist-to-height ratio (0.5) were used to characterise abdominal obesity [38,39].

### 2.3. Dietary Intake

A 24 h recall questionnaire was used to gather dietary data [40]. Participants were questioned by trained ELS field workers about their food consumption the previous day. All of the subjects’ parents or caregivers were questioned about their nutritional consumption during the preceding 24 h. Using a pre-tested questionnaire and food models that represented typical serving sizes of regional meals, the estimated portion sizes of the items ingested were recorded in as much detail as feasible. Each participant’s food consumption was recorded for an average of two days. Two food intakes—one for the week and the other for the weekend—were recorded [41]. This is due to the fact that what people eat during the week varies from what they eat on the weekends. Particularly on Saturdays, people tend to consume significantly more on the weekends than they do during the week [42]. The SAFOODS and local food tables were used to analyse the average sodium and potassium intakes [43], which were then compared to the recommended intakes for each nutrient as stated in the Consensus Study Report (2019) [44]. The WHO suggests using the sodium-to-potassium ratio of 1:1 (or 1) to determine the average molar sodium-to-potassium ratio [12]. The sodium and potassium intakes in this study were translated from milligrams to millimoles, since the suggested sodium/potassium ratio is expressed in moles. The molar sodium/potassium ratio was calculated using the calculation shown below [45]:23 mg sodium = 1 mmol sodium;
39 mg potassium = 1 mmol potassium

### 2.4. Statistical Analysis

In order to conduct the statistical analysis, the IBM SPSS Statistics software package (version 27.0) was employed. Descriptive statistics were produced for all the variables to provide frequencies (expressed as percentages), means, and standard deviations in order to describe and characterise the samples. The normality of the variables was evaluated by the Shapiro–Wilk test. To ascertain the variation in males and females throughout the years and between the years, parametric (one-way ANOVA), Chi-square, and Fisher’s exact tests were used. To evaluate the relationship between sodium and potassium consumption, as well as their ratio with BP, WC, and WHtR, and further analyse the relationship with hypertension and abdominal obesity, generalised estimating equations (linear and binary logistics) were used. A *p*-value of ≤0.05 was chosen as the statistical significance for all tests.

## 3. Results

### 3.1. Characteristics of the Population

Table 1 indicates the characteristics of the population for each year stratified by sex, with a total of 325 participants. Although the dietary mean of Na consumption increased in both males and females with time, the rise was not significant in either males (*p* > 0.05) or females (*p* > 0.05). The dietary mean of K intake significantly decreased from the first year of assessment to the last year of assessment in both males (*p* < 0.001) and females (*p* < 0.001). Meanwhile, the dietary mean of the Na/K ratio significantly increased over the years in both males (*p* < 0.001) and females (*p* < 0.001). Furthermore, the mean of both SBP and DBP significantly increased from the first year of assessment to the last year of assessment, respectively, in both males (*p* < 0.001) and females (*p* < 0.001). The mean of abdominal indices (WC and WHtR) significantly increased over the years in both males (*p* < 0.001) and females (*p* < 0.001). Table 2 further shows that the dietary mean of the micro-nutrients (Na (*p* < 0.05), K intake (*p* < 0.001), and Na/K ratio (*p* < 0.001)) significantly increased over the years for the total population. In addition, the mean of BP measurements of the total population significantly increased from the first point of measurement, with SBP (*p* < 0.001) increasing to 120.75 mmHg and DBP (*p* < 0.001) increasing to 70.67 mmHg in the last year of measurement, respectively. The mean of abdominal indices significantly increased from the first measurement (54.44 cm-WC and 0.41-WHtR), with WC increasing to 79.0 cm and WHtR increasing to 0.48 in the last year.

Table 3 presents the most frequent food items in the diets of the Ellisras participants from 2001 and 2015. In 2001, maize or sorghum porridge was the most stable food among the Ellisras children, with jam being the least used item. In 2015, fried chicken with skin was the most used item, and pilchards were the least used food item among the Ellisras young adults.

### 3.2. The Prevalence of Sodium, Potassium, Hypertension and Abdominal Obesity

Table 4 indicates a significant increase in the prevalence of hypertension from the first year to the third year in both males and females, respectively (*p* < 0.001). The prevalence of abdominal obesity according to WC significantly decreased in males, whereas in females, it increased from the first point of measurement to the third year (*p* < 0.001). In addition to this, both males’ and females’ prevalence of abdominal obesity according to WHtR significantly increased over the measured years (*p* < 0.001). The results in Table 5 show the prevalence of hypertension was 4.9% at the first measurement, and it significantly increased to 23.4% in the last year of assessment (*p* < 0.001). In addition, the prevalence of abdominal obesity according to WC significantly decreased by 5% from the first point of measurement to the last year of assessment (*p* < 0.001). Meanwhile, the prevalence of abdominal obesity significantly increased from 0.3% at the first measurement to 35.7% in the last year of measurement (*p* < 0.001).

### 3.3. The Association of Sodium Intake, Potassium Intake and Sodium-to-Potassium Ratio with Dependent Variables

The results in Table 6 show a significant positive association between potassium intake and WHtR (β = 0.019, (95% CL: 0.004, 0.034) *p*-value = 0.012), and even after age and gender were considered in the model, there was still an association with WHtR (β = 0.018, (95% CL: 0.002, 0.034) *p*-value = 0.024). The sodium-to-potassium ratio was significantly associated with SBP (β = 4.326, (95% CL: 2.056, 6.595) *p*-value < 0.001), DBP (β = 2.028, (95% CL: 0.703, 3.353) *p*-value = 0.003), WC (β = 4.191, (95% CL: 2.080, 6.302) *p*-value < 0.001) and WHtR (β = 0.014, (95% CL: 0.003, 0.026) *p*-value = 0.015), respectively. However, when the model was adjusted for age and sex, an association was not found between the sodium-to-potassium ratio and blood pressure measurements and abdominal indices, respectively. Table 5 shows that in this study, it was further observed that the Na/K ratio increased the risk of developing hypertension (Exp β = 1.603, (95% CL: 1.164, 2.207) *p*-value = 0.004) and abdominal obesity (Exp β = 1.797, (95% CL: 1.207, 2.677) *p*-value = 0.004) compared to Na and K intake alone. However, when age and sex were taken into account, the sodium-to-potassium ratio was not found to increase the risk of developing hypertension and abdominal obesity, as shown in Table 7.

## 4. Discussion

The purpose of this study was to investigate the association of sodium and potassium intake and their ratio on hypertension and abdominal obesity in the same participants over time in Ellisras. The results of our study indicated that K was significantly associated with WHtR. The outcome of these results could be due to a lack of intake of fruits and vegetables in 2015 compared to 2001 (e.g., the sample population consumed bananas and oranges, which are high in potassium), resulting in a decline in the mean intake of K over the years. Most studies in this field that have been conducted over the past five years indicate that the K effect on obesity is a new topic, and more robust studies with better design are warranted [46], especially on abdominal obesity, as previous studies mostly focused on BMI and not abdominal indices such as WC and WHtR. It is unknown exactly how potassium consumption affects obesity/MetS. Central obesity is a component of metabolic syndrome, and the mechanisms of obesity and MetS are homogeneous. Potassium channel function and obesity are associated [47,48]. Our results further showed a significant association between Na/K ratio and both BP measurements (increasing SBP by 4.236 mmHg and DBP by 2.028 mmHg) and abdominal indices (increasing WC by 4.191 cm and WHtR by 0.0014 cm); the association was positive. Despite the lack of studies that could be compared to this study, Pereira et al. (2019) found independent associations of Na and K intake with BP, but when evaluated in a combined manner, as in the case of the Na/K ratio, the effect was potentiated [49].

Our results, on the one hand, showed no relationship between Na and BP measurements and abdominal indices. Although Na mean intake significantly increased over time, the increase was not significant enough to be associated with BP measurement and abdominal indices in this sample population of Ellisras.

Our study results revealed that the Na/K ratio is the predictor of HT and abdominal obesity compared to Na and K alone. This means that an increase in the Na/K ratio increases the risk of developing hypertension by 1.603-fold and abdominal pain by 1.797-fold over time. Due to the lack of sufficient studies, we could not compare this study with relevant studies. However, Ge et al. (2016) also discovered that the urine Na/K ratio was independently related to obesity and that a high Na/K ratio could increase the risk of obesity [50]. However, no link was discovered between the Na/K ratio (as determined by self-reporting) and obesity [51]. This contradicts the findings of this study, as the Na/K ratio was calculated using self-reported dietary data in both groups. Because the topic is new, data on the relationship between Na/K and abdominal obesity are inconsistent. On the other hand, our study is consistent with the literature, which states that the Na/K ratio outperforms Na and K, as individual predictors of BP change in various investigations. Most of the studies conducted in hypertensive patients [23] indicated that a higher Na/K ratio may lead to higher BP during follow-up [52]. Elevated levels of Na intake and inadequate levels of K intake may affect the development of hypertension [12]. To further agree with this literature, our results indicated an increase in Na over the years and a decrease in K over the years. In all the years, there were significant mean differences in males and females for the intake of potassium and the sodium-to-potassium ratio. In addition, there were significant mean differences in abdominal measurements for both males and females over the years.

On the other hand, BP measurements showed a significant mean difference in males and females over the years. Significant differences in the prevalence of hypertension, WC, and WHtR were observed in both genders over the years. Over time, there were significant variations in potassium, the sodium-to-potassium ratio, hypertension, WC, and WHtR prevalence. It must be noted that many of the participants were classified as underweight, as stated by van Den Ende et al. (2014) [31]. Therefore, the high prevalence of abdominal obesity according to WC compared to the prevalence of abdominal obesity according to WHtR 2001 might be due to bloating from malnourishment rather than visceral fat [41]. There was no significant change in the prevalence of sodium over time.

The change in political, social, and economic factors in South Africa has resulted in increased urbanisation and progress [31]. The increased accessibility, availability, and affordability of processed foods in South Africa are of concern, as these types of foods are generally considered to be high in fat, sugar or salt (sodium) [53,54]. Although our study found a lower intake of sodium and potassium, this is probably due to the fact that it was conducted in a rural settlement, because excessive intake of sodium and a deficient potassium intake may result in health issues.

There are a few limitations regarding this study. The first is the use of the 24 h recall questionnaire compared to the 24 h urinary excretion, as 24 h excretion is considered the golden standard method of obtaining data on sodium and potassium intake in population surveys and is more accurate than the 24 h recall questionnaire [30]. The second limitation is the small sample size and the lack of different ethnicities from different geographical regions. Future sodium excretion data over time, from childhood to adulthood within these regional areas, will be ideal for evaluating post-legislation salt intakes and their impact on public health [55]. Third, it does not include the socio-economic status of the sample population. In addition, the gap between the years is a limitation of this study. This was due to financial constraints. The strength of this study is the use of longitudinal data. The current study is valuable and informative regarding the status of sodium and potassium intake in a sample of South Africans. Tracking the dietary habits of children into adulthood is vital, as children with extremely high levels of sodium intake tend to maintain those levels for some time [56]. These could lead to the development of hypertension and abdominal obesity. Thus, the close monitoring of children is needed for better management of their health.

## 5. Conclusions

In conclusion, this current study showed a significant positive association between K and WHtR. In addition, there is a significant positive relationship between the Na/K ratio, BP measurements (SBP and DBP), and abdominal indices (WC and WHtR). An increase in the Na/K ratio was further found to increase the risk of developing HT and abdominal obesity over time. However, more data are crucial in establishing the effectiveness of Na reduction and the increase in K intake, and with the use of more valuable methods, we can better understand their effect on blood pressure and abdominal weight, especially in Africa. Longitudinal studies in children provide a tremendous global resource to direct prevention strategies for HT and abdominal obesity.

## Figures and Tables

**Figure 1 children-10-01330-f001:**
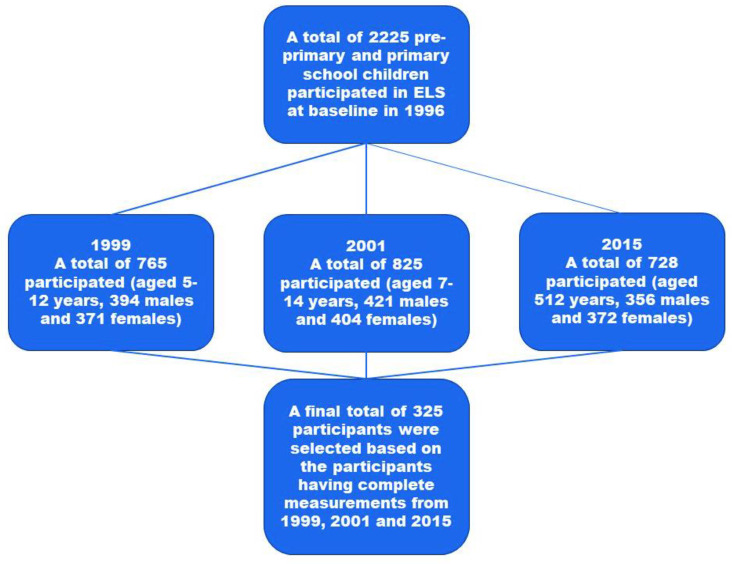
The flow chart of the study participants over the years.

**Table 1 children-10-01330-t001:** Characteristics of the population over the years, stratified by sex.

Year	1999		2001		2015	
	Males (*n* = 161)	Females (*n* = 164)	Males (*n* = 161)	Females (*n* = 164)	Males (*n* = 161)	Females (*n* = 164)
**Variables**	Mean ± SD	Mean ± SD	Mean ± SD	Mean ± SD	Mean ± SD	Mean ± SD
**Age**	8.78 ± 1.86	9.59 ± 1.75	10.78 ± 1.86	11.54 ± 1.75	24.79 ± 1.84	25.52 ± 3.13
**Na (mg/d)**	482.06 ± 593.49	532.84 ± 593.03	482.06 ± 593.49	535.82 ± 592.44	661.45 ± 988.80	717.18 ± 1082.93
**K (mg/d)**	1043.97 ± 836.65	1165.77 ± 963.26	1043.97 ± 836.65	1175.37 ± 968.15	756.28 ± 657.31 **	753.27 ± 678.56 **
**Na/K ratio (mmol/d)**	0.90 ± 1.03	0.89 ± 0.93	0.90 ± 1.03	0.89 ± 0.92	2.03 ± 2.75 **	1.92 ± 2.67 **
**SBP (mmHg)**	99.86 ± 11.71	102.17 ± 11.43	95.51 ± 9.90	99.39 ± 11.21	126.72 ± 13.11 **	115.00 ± 10.63 **
**DBP (mmHg)**	61.90 ± 9.89	62.92 ± 9.58	63.58 ± 7.21	64.66 ± 8.04	71.45 ± 10.17 **	69.93 ± 8.99 **
**WC (cm)**	53.93 ± 3.77	54.96 ± 4.69	56.65 ± 4.14	58.23 ± 4.30	74.53 ± 9.12 **	84.01 ± 15.39 **
**WHtR**	0.41 ± 0.02	0.40 ± 0.02	0.41 ± 0.04	0.41 ± 0.04	0.43 ± 0.08 **	0.52 ± 0.12 **
**Height**	130.36 ± 120.08	135.10 ± 10.68	138.51 ± 11.68	143.47 ± 10.77	173.54 ± 12.83 **	162.79 ± 10.29 **

*n*—number of individuals. Na—sodium, K—potassium, Na/K ratio—sodium/potassium ratio, SBP—systolic blood pressure, DBP—diastolic blood pressure, WHtR—waist-to-height ratio, mg/d—milligram per day, mmol/d—millimole per day, cm—centimetre, ** *p*-value < 0.001.

**Table 2 children-10-01330-t002:** Comparison of means of sodium, potassium intake, sodium-to-potassium ratio, systolic and diastolic blood pressure over the years.

Year	1999	2001	2015	*p*-Value
**Variables**	Mean ± SD	Mean ± SD	Mean ± SD	
**Na (mg/d)**	507.69 ± 592.89	509.27 ± 592.66	689.84 ± 1036.56	0.008 *
**K (mg/d)**	1105.43 ± 903.43	1110.48 ± 906.59	754.75 ± 667.16	0.000 **
**Na/K ratio (mmol/d)**	0.89 ± 0.98	0.89 ± 0.98	1.97 ± 2.70	0.000 **
**SBP (mmHg)**	101.02 ± 11.61	97.47 ± 10.74	120.75 ± 13.26	0.000 **
**DBP (mmHg)**	62.41 ± 9.73	64.12 ± 7.64	70.67 ± 9.60	0.000 **
**WC (cm)**	54.44 ± 4.27	57.47 ± 4.30	79.40 ± 13.53	0.000 **
**WHtR**	0.41 ± 0.24	0.41 ± 0.04	0.48 ± 0.11	0.000 **

Na—sodium, K—potassium, Na/K ratio—sodium/potassium ratio, SBP—systolic blood pressure, DBP—diastolic blood pressure, WHtR—waist-to-height ratio, SD—standard deviation, mg/d—milligram per day, mmol/d—millimole per day, cm—centimetre, * *p*-value < 0.05; ** *p*-value < 0.001.

**Table 3 children-10-01330-t003:** The most frequently used food items for the years 2001 and 2015 in Ellisras, from the most used to the least.

2001	2015 (%)
Maize porridge or Sorghum	Fried chicken with skin (23.8)
Sugar (white)	Pap (22.6)
Homemade bread	Cold drink (16.9)
Chicken	White sugar (14)
Spinach	Vetkoek (5.8)
Non-dietary creamer	Fried beef (4.7)
Beef	Peanut butter (4.4)
Red meat (from goats and wild animals)	Samp (2.6)
Tomato and onion	Yoghurt (2.4)
Cooked dry beans	Spinach (2.0)
Cold drink (mostly Coke)	Pilchards (0.5)
Peanut butter	
Sweets	
Mashontja (Mopani worms)	
Bananas and oranges	
Cow milk	
Jam	

Source adapted [18,19].

**Table 4 children-10-01330-t004:** The prevalence of sodium, potassium, hypertension and abdominal obesity stratified by sex.

Year	1999		2001		2015	
**Dependent variables**	Males *n* (%)	Females *n* (%)	Males *n* (%)	Females *n* (%)	Males *n* (%)	Females *n* (%)
**Na (mg/d) above the adequate intake**	14 (8.7)	18 (11.0)	20 (12.4)	15 (9.1)	19 (11.8)	17 (10.4)
**K (mg/d) below the adequate intake**	156 (96.9)	153 (93.3)	153 (95.0)	150 (91.5)	155 (96.3)	163 (99.4) *
**Na/K (mmol/d) above the recommended intake**	57 (35.4)	56 (34.1)	58 (36.0)	56 (34.1)	84 (52.2) **	82 (50.0) *
**Hypertension**	8 (5.0)	8 (4.9)	4 (2.5)	6 (3.7)	36 (22.5) **	40 (24.2) **
**Abdominal obesity according to WC**	58 (36.0)	71 (43.6)	108 (67.1)	131 (79.9)	17 (10.6) **	96 (58.2) **
**Abdominal obesity according to WHtR**	0 (0)	1 (0.6)	4 (2.5)	10 (6.1)	19 (11.9) **	97 (58.8) **

*n*—number of individuals. Na—sodium, K—potassium, WC—waist circumference, WHtR—waist-to-height ratio, mg/d—milligram per day, mmol/d– millimole per day, * *p*-value < 0.05, ** *p*-value < 0.001.

**Table 5 children-10-01330-t005:** The prevalence of sodium, potassium, hypertension and abdominal obesity over the years.

Year	1999	2001	2015	Chi-Square *p*-Value
**Dependant variables**	*n* (%)	*n* (%)	*n* (%)	
**Na (mg/d) above the adequate intake**	32 (9.8)	35 (10.8)	36 (11.1)	0.868
**K (mg/d) below the adequate intake**	309 (95.1)	303 (93.2)	318 (97.8)	0.019 *
**Na/K (mmol/d) above the recommended intake**	113 (34.8)	114 (35.1)	166 (51.1)	<0.001 *
**Hypertension**	16 (4.9)	10 (3.1)	76 (23.4)	<0.001 **
**Abdominal obesity according to WC**	129 (39.8)	239 (73.5)	113 (34.8)	<0.001 **
**Abdominal obesity according to WHtR**	1 (0.3)	14 (4.3)	116 (35.7)	<0.001 **

*n*—number of individuals. Na—sodium, K—potassium, WC—waist circumference, WHtR—waist-to-height ratio, mg/d- milligram per day, mmol/d- millimole per day, * *p*-value < 0.05, ** *p*-value < 0.001.

**Table 6 children-10-01330-t006:** The association of sodium intake, potassium intake and sodium-to-potassium ratio with blood pressure and abdominal indices using GEE (linear).

	Na (mg/d)			K (mg/d)			Na/K (mmol/d)		
**Unadjusted**	**β**	**95% Cl**	***p*-value**	**β**	**95% Cl**	***p*-value**	**β**	**95% Cl**	***p*-value**
**SBP**	−0.507	−4.379, 3.366	0.798	2.033	−2.022, 6.088	0.326	4.326	2.056, 6.595	<0.001 **
**DBP**	−0.240	−2.287, 1.807	0.818	0.736	−1.738, 3.211	0.560	2.028	0.703, 3.353	0.003 *
**WC**	−0.735	−4.071, 2.600	0.666	2.314	−1.017, 5.644	0.173	4.191	2.080; 6.302	<0.001 **
**WHtR**	−0.001	−0.016, 0.013	0.866	0.019	0.004, 0.034	0.012 *	0.014	0.003, 0.026	0.015 *
**Adjusted for age and sex**									
**SBP**	0.915	−1.769, 3.599	0.504	0.579	−2.662, 3.821	0.726	0.967	−0.655, 2.589	0.243
**DBP**	0.311	−1.571, 2.194	0.746	0.283	−2.142, 2.709	0.819	0.798	−0.418, 2.014	0.198
**WC**	1.052	−0.668, 2.773	0.231	1.247	−0.642, 3.136	0.196	0.469	−0.701, 0.617	0.432
**WHtR**	0.004	−0.009, 0.017	0.57	0.018	0.002, 0.034	0.024 *	0.005	−0.005, 0.015	0.329

Na—sodium, K—potassium, Na/K ratio—sodium/potassium ratio, SBP—systolic blood pressure, DBP—diastolic blood pressure, WC—waist circumference, WHtR—waist-to-height ratio, mg/d—milligram per day, mmol/d—millimole per day, β—beta coefficient, * *p*-value < 0.05, ** *p*-value < 0.001.

**Table 7 children-10-01330-t007:** The risk measure of sodium, potassium intake and sodium-to-potassium ratio causing the development of hypertension and abdominal obesity according to WC and WHtR using GEE (binary logistic).

	Na (mg/d)			K (mg/d)			Na/K (mmol/d)		
**Unadjusted**	**Exp (β)**	**95% Cl**	***p*-value**	**Exp (β)**	**95% Cl**	***p*-value**	**Exp (β)**	**95% Cl**	***p*-value**
**Hypertension**	0.959	0.590, 1.558	0.865	1.941	0.762, 4.941	0.164	1.603	1.164, 2.207	0.004 *
**Abdominal obesity according to WC**	1.131	0.730, 1.754	0.581	0.634	0.321, 1.254	0.190	0.912	0.684, 1.216	0.529
**Abdominal obesity according to WHtR**	1.066	0.579, 1.964	0.837	1.964	0.589, 6.545	0.272	1.797	1.207, 2.677	0.004 *
**Adjusted for age and sex**									
**Hypertension**	0.873	0.530, 1.437	0.593	0.679	0.283, 1.632	0.387	0.968	0.668, 1.403	0.864
**Abdominal obesity according to WC**	1.136	0.715, 1.805	0.588	0.698	0.362, 1.346	0.283	0.989	0.731, 1.338	0.943
**Abdominal obesity according to WHtR**	0.654	0.312, 1.372	0.262	0.702	0.221, 2.233	0.549	0.781	0.493, 1.239	0.295

Na—sodium, K—potassium, Na/K ratio—sodium/potassium ratio, WC—waist circumference, WHtR—waist-to-height ratio, mg/d—milligram per day, mmol/d—millimole per day, Exp β—exponential beta, * *p*-value < 0.05.

## Data Availability

Data will be made available upon request.

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
