# Peer review of "A Longitudinal Investigation on the Effects of Sodium and Potassium Intake on the Development of Hypertension and Abdominal Obesity from Childhood to Young Adulthood amongst Ellisras Rural Population, South Africa"

_children, 2023, doi:10.3390/children10081330_

Round 1
Reviewer 1 Report
The manuscript aims to evaluate a longitudinal study of interactions between Na, K intake, Na/K ratio and changes in blood pressure and parameters of abdominal obesity in the Ellisras region of South Africa. The authors of the manuscript in this study determined it using measurements of raw Na and K calculations from a 24-hour nutritional protocol, and anthropometric indicators and blood pressure were measured by standard methods, in children evaluated by percentiles in adults according to WHO recommendations. Introduction - in line 61 - 2 dots at the end of the sentence - the quality of the introduction would be further enhanced by citations from the last 5 years. Material and methods - in section 2.1. it would be appropriate to supplement the data on the full-term birth weight of the children - whether it was normal or subnormal (of course, if the authors of the study have this information) - it is also an important factor affecting the development of obesity. - line 73 - it would be appropriate to edit Figure 1 graphically, the text is partially blurred, it may be difficult for readers to read Results - line 127 - it would be appropriate to partially reformulate the sentence - line 144 - table header 1A - The first letter in the word Characteristic should be capitalized - in table 1A – for Na, K, and Na/K, the unit is mg; mg; mmol and in the legend below the table the unit mg/d and mmol/d is indicated, it would be appropriate to unify the use of units - in my opinion, it is more appropriate to use mg per day; mmol per day - similarly in Table 1B - line 153-157 - the text paragraph is shifted to the right - table 2 - column for the year 2001 - next to the consumed meals, indicate %, as it is in the column for the year 2015 - lines 170-182 - the entire paragraph of text is shifted to the right - line 172, modify the wording of the sentence slightly - in Tables 3A; 3B unify the units as Table 1A, 1B - lines 190-207 - the entire paragraph of text is shifted to the right - In the header of Tables 4 and 5, the units for Na; K and Na/K are missing Discussion - the quality of the discussion would be further enhanced by citations from the last 5 years. Conclusion - I have not comments. References In the manuscript were used 41 literary sources (including 4 self-citations) of which only 5 sources are from the last 5 years; 16 literary sources are from the period of 5 to 10 years and 20 sources are older than 10 years. Therefore, it would be advisable to upgrade some sources.
minimal editing is needed - see my comments
Reviewer 2 Report
Here are my comments on the manuscript by Raphadu et al. entitled “A longitudinal investigation on the effects of sodium and potassium intake have on the development of hypertension and abdominal obesity from childhood to young adulthood amongst Ellisras rural population, South Africa”
-In the abstract section, please add the method for the sodium and potassium determination, and the age of the groups studied. In addition, define the acronym WHtR.
--The introduction section should include statistics on hypertension and obesity of the population of South Africa.
-According to the next phrase: “Several studies have indicated a significant association between sodium and obesity 49 among adults [3, 6-7].” The references 3 and 6 do not contain information about the association between sodium and obesity. Please check these references and add the appropriate ones.
-In the following phrase: “South Korea, the UK and Australia also identified a significant positive association between sodium and obesity among both the youth and adults [3; 8-9].” The reference 3 and 9 have no relation to obesity. Please add the appropriate references.
-In the following phrase: Blood pressure was measured using an Omron M2 blood pressure device as described by Whelton et al. (2018). The authors should add the reference 16 at the end of phrase.
-The authors should add a supplemental table with reported sodium and potassium percentages values for the most common food intake by the South Africa population studied.
-According to the next phrase: “The mean of abdominal indices (WC and WHtR) significantly increased over the years, in both males (p< 0.001) and females (p< 0.001).” With respect to WC and WHtR, the results are as expected because the children grow and gain body mass. I suggest to include in this study the body mass index (BMI).
Reviewer 3 Report
Numerous studies available in the Literature (performed on large cohorts of adults or children/young people) revealed that excessive sodium (Na) and inadequate potassium (K) intakes are associated with an increased risk of high blood pressure (BP) and higher BMI, which are the primary risk factors for cardiovascular diseases.
Please see Libuda L et al., 2012, Doi: 10.1017/S1368980011002138; Murakami K et al., 2015, Doi: 10.1017/S0007114515000495; Sahar Golpour-Hamedani et al., 2022, Doi: 10.1186/s12937-022-00776-y; Tali Elfassy et al., 2018, Doi: 10.1002/oby.22089
The sodium—obesity relationship has been attributed to indirect downstream processes related to increased energy intake. However, emerging evidence suggests a direct correlation between sodium and obesity exists independent of energy intake. In addition, a salty diet may cause a greater consumption of sugar-sweetened beverages, which are associated with weight gain, while K can influence carbohydrate accumulation and glucose homeostasis.
In this regard, the manuscript prepared by Thato Tshepo Raphadu et al. investigates the effect of sodium and potassium intake and their ratio on hypertension and abdominal obesity from childhood into young adulthood in Ellisras. Although associated with a statistical result, the study does not provide significant new data in this area of research.
Reviewer 4 Report
1. The introduction does not provide a comprehensive background. Needs to be elaborated further.
2. The resolution quality of Figure 1 is not good, it must be improved.
3. The presentation of the table is not good, it must be adjusted to the journal format and the separation of tables 1 a and 1 b is not quite right. It should be table 1 and table 2.
4. The English version now has to be extensively improved and needs to be proofread and edited by Native Speakers; the certificate must be attached in later revisions.
The English version now has to be extensively improved and needs to be proofread and edited by Native Speakers; the certificate must be attached in later revisions.
Round 2
Reviewer 2 Report
I have no comments.
The English language is fine, but minor editing of English language required.
Author Response
Comments on the Quality of English Language
Point 1:
The English language is fine, but minor editing of English language required.
Response 1: The manuscript has been submitted for proofreading and editing to improve the English Language, thank you.
Reviewer 3 Report
The authors improved the manuscript, by pointing out the relevance of their research. I have only a minor suggestion for the authors:
- To improve the readability of the text associated with Figure 1, please use bold or change the size of the text related to it.
Author Response
Point 1:
The authors improved the manuscript, by pointing out the relevance of their research. I have only a minor suggestion for the authors:
- To improve the readability of the text associated with Figure 1, please use bold or change the size of the text related to it.
Response 1: Bold and a sight change of the font size has been applied as suggested, thank you.
Reviewer 4 Report
Not enough revision and the certificate of english must be attached in the response to reviewers
Certificate?
Author Response
Point 1: Not enough revision and the certificate of English must be attached in the response to reviewers
Response 1: We apologize the certificate was attached together with the revised manuscript and some new information has been added to the introduction.
Please see the attachment
